

# Evaluating the diurnal cycle in cloud top temperature from SEVIRI

Sarah Taylor[1], Philip Stier[1], Bethan White[1], Stephan Finkensieper[2], and Martin Stengel[2]

[1]Department of Physics, University of Oxford, UK
[2]Deutscher Wetterdienst (DWD), Offenbach, Germany

*Correspondence to:* S. A. Taylor (sarah.taylor@physics.ox.ac.uk)

**Abstract.** The variability of convective cloud spans a wide range of temporal and spatial scales and is of fundamental importance for global weather and climate systems. Datasets from geostationary satellite instruments such as SEVIRI provide high time resolution observations across a large area. In this study we use data from SEVIRI to quantify the diurnal cycle of cloud top temperature within the instrument's field of view and discuss these results in relation to retrieval biases.

We evaluate SEVIRI cloud top temperatures from the new CLAAS-2 (CLoud property dAtAset using SEVIRI, Edition 2) dataset against CALIOP data. Results show a mean bias of + 0.44 K with a standard deviation of 11.7 K, which is in agreement with previous validation studies. Analysis of the spatiotemporal distribution of these errors shows that absolute retrieval biases vary from less than 5 K over the southeast Atlantic Ocean up to 30 K over central Africa at night. Night and daytime retrieval biases can also differ by up to 30 K in some areas, potentially contributing to biases in the estimated amplitude of the diurnal

cycle. This illustrates the importance of considering spatial and diurnal variations in retrieval errors when using the CLAAS-2 dataset.

     Keeping these biases in mind, we quantify the seasonal, diurnal and spatial variation of cloud top temperature across SEVIRI's field of view using the CLAAS-2 dataset. By comparing the mean diurnal cycle of cloud top temperature with the retrieval bias we find that diurnal variations in the retrieval bias can be small, but are often of the same order of magnitude as

the amplitude of the observed diurnal cycle, indicating that in some regions the diurnal cycle apparent in the observations may be significantly impacted by diurnal variability in the accuracy of the retrieval.

     We show that the CLAAS-2 dataset can measure the diurnal cycle of cloud tops accurately in regions of stratiform cloud such as the southeast Atlantic Ocean and Europe, where cloud top temperature retrieval biases are small and exhibit limited spatial and temporal variability. Quantifying the diurnal cycle over the tropics and regions of desert is more difficult, as retrieval

biases are larger and display significant diurnal variability. CLAAS-2 cloud top temperature data are found to be of limited skill in measuring the diurnal cycle accurately over desert regions. In tropical regions such as Central Africa, the diurnal cycle can be described by the CLAAS-2 data to some extent, although retrieval biases appear to reduce the amplitude of the real diurnal cycle of cloud top temperatures.

     This is the first study to relate the diurnal variations in SEVIRI retrieval bias to observed diurnal cycles in cloud top temper-

ature. Our results may be of interest to those in the observation and modelling communities when using cloud top properties data from SEVIRI, particularly for studies considering the diurnal cycle of convection.



## 1  Introduction

Convection is one of the core building blocks of tropical weather and climate, playing important roles in large-scale atmospheric circulations, the hydrological cycle, the global energy budget and the transport of heat, moisture, momentum and aerosols throughout the troposphere (Grabowski and Petch, 2009). The temporal and spatial variability of convective cloud are therefore
of fundamental importance for global weather and climate systems.

Spatial scales of convection range from thousands of kilometers for mesoscale convective systems (MCS) to a few kilometers for individual convective plumes, while time scales of convective variability range from minutes through to seasons. In particular, the diurnal and seasonal cycles of tropical convection, driven by variations in solar forcing, are among the strongest and most fundamental modes of variation in the global weather and climate systems.

A large number of observational studies using data from rain gauges (Wallace, 1975; Gray and Jacobsen, 1977), surface weather reports (Dai, 2001) and both polar-orbiting (Nesbitt and Zipser, 2003; Yang et al., 2008; Stratton and Stirling, 2012) and geostationary (Meisner and Arkin, 1987; Janowiak et al., 1994; Chen and Houze, 1997; Yang and Slingo, 2001; Schröder et al., 2009) satellites have attempted to quantify the diurnal cycle of convection over land. These studies found an early afternoon maximum in convective precipitation, followed by a minimum in cloud top temperature approximately three hours
later. The two features are thought to correspond to the beginning and end of the mature stage of convection (Schröder et al., 2009).

However, these large-scale features, driven by insolation, display regional and seasonal variations (Yang and Slingo, 2001; Schröder et al., 2009) and can be overridden by other factors such as orography (Yang and Slingo, 2001; Nesbitt and Zipser, 2003; Vondou et al., 2010), land-sea breezes (Chen and Houze, 1997; Yang and Slingo, 2001; Halladay et al., 2012) and the
organisation of convection (Nesbitt and Zipser, 2003).

The amplitude of the diurnal cycle of convection is smaller over the ocean than over land (Harrison et al., 1988), due to the ocean's higher heat capacity, and because ocean mixing distributes incoming solar radiation away from the surface. Most studies show a substantial pre-dawn peak in convective cloud over the oceans (Janowiak et al., 1994; Yang and Slingo, 2001; Nesbitt and Zipser, 2003; Bain et al., 2010; Stengel et al., 2014). The mechanisms responsible for this overnight peak in
convective cloud remain uncertain (Bain et al., 2010), but are thought to be related to atmospheric instability caused by night time radiative cooling (Randall et al., 1991) and to the presence of a larger number of MCSs during the night (Chen and Houze, 1997; Nesbitt and Zipser, 2003).

However, general circulation and numerical weather prediction models which parameterise convective processes fail to capture the observed diurnal cycle of convective cloud (Yang and Slingo, 2001; Guichard et al., 2004; Grabowski et al., 2006;
Stratton and Stirling, 2012). This is generally the result of convection initiating shortly after sunrise, which develops too rapidly, quickly reaching the tropopause and producing precipitation (Guichard et al., 2004; Stratton and Stirling, 2012).

Studies by Guichard et al. (2004); Grabowski et al. (2006) and Sato et al. (2009) show that in some cases, cloud resolving models (CRMs), which explicitly resolve convection, are capable of correctly predicting the amplitude and phase of the diurnal cycle in convection. Over land, this accuracy is strongly dependent on horizontal resolution, requiring grid lengths of around





km (Guichard et al., 2004), or 500 m (Grabowski et al., 2006). Over the ocean, horizontal resolution appears to be less important, which is likely due to differences in predominant cloud types and lifecycles (Sato et al., 2009).

Observations can both improve our theoretical understanding and provide a useful test of a model's ability to capture the various scales of variation of convective cloud. While low Earth orbit satellites can provide observations at high spatial resolution,
their temporal sampling is limited. Polar-orbiting satellites in sun-synchronous orbit, such as those in the A-Train constellation of satellites, observe any given point (except polar regions, which are observed more often) no more than twice per day and always at the same local solar time. Other low Earth orbit satellites, such as the Tropical Rainfall Measuring Mission (TRMM), are able to sample a given point at varying local solar times, thereby providing statistical diurnal cycle observations. Individual low Earth orbit satellites are therefore unable to observe the temporal evolution of convective clouds.

In comparison, while the spatial resolution of geostationary satellites is limited, they provide high temporal resolution observations over a large area. The Spinning Enhanced Visible and Infrared Imager (SEVIRI) instrument on board the geostationary Meteosat Second Generation (MSG) satellites has a spatial resolution of 3 km at the satellite nadir and a temporal resolution of 15 minutes, with a record of observations reaching back to 2004. Its field of view, hereafter referred to as the SEVIRI 'disc', covers the entire continents of Africa and Europe, as well as the Middle East, eastern South America, and both the
Atlantic Ocean and the western part of the Indian ocean. The continuous nature of SEVIRI's observations makes them ideal for investigating the temporal and spatial variability of cloud across a large area and period of time.

It should be noted that passive imagers such as SEVIRI observe the radiometric height of the cloud. This differs from the physical cloud top height which can be measured very accurately by active lidar instruments such as the Cloud-Aerosol Lidar with Orthogonal Polarization (CALIOP) even for optically thick cloud (Sherwood et al., 2004). SEVIRI is therefore expected
to underestimate cloud top pressure (CTP) and cloud top height (CTH), and overestimate cloud top temperature (CTT) relative to CALIOP. This is due to differences in the properties observed by the two instruments, rather than an error in the SEVIRI retrieval.

Although it is often assumed that optically thick clouds have sharp boundaries and can be expected to radiate as black bodies (Sherwood et al., 2004; Stubenrauch et al., 2013), the radiometric height of a cloud may be several kilometers below
physical cloud top height depending on its extinction profile at the cloud top and its vertical size (Stubenrauch et al., 2013). In particular, glaciated clouds tend to have poorly defined edges, even when convectively active (Sherwood et al., 2004), and so optical depths increase slowly with distance from the cloud top (Stubenrauch et al., 2013).

Sherwood et al. (2004) found that radiometric cloud tops retrieved from the Geostationary Operational Environmental Satellite 8 (GOES-8) were on average 1 km below, or 5-7 K above, the visible cloud tops observed by NASA's Cloud Physics Lidar
(CPL). This bias increased to 2 km for the highest cloud tops, and was not found to vary with cloud albedo. Other studies (Heymsfield et al., 1991; Minnis et al., 2008; Stubenrauch et al., 2010, 2013) show similar biases of between 0.5 and 3 km for high clouds in the tropics.

An additional cause of differences between these the cloud properties observed by CALIOP and SEVIRI is the occurrence of optically thin clouds, or cloud layers. These clouds can usually be detected by CALIOP. Their impact on passively measured
radiation is however very small and thus not properly detected by passive imager sensors. This is a particular issue in the case





of pixels containing semi-transparent cloud types, which have very low optical depths, as it is difficult to fully account for contributions from surface radiation, or low cloud layers underneath (Smith and Platt, 1978). SEVIRI is also not sensitive to clouds with very low optical depths which can be detected by lidar (Heidinger and Pavolonis, 2009; Stubenrauch et al., 2010; SAFNWC/MSG, 2012).

In this study we evaluate SEVIRI CTT retrievals dataset against data from CALIOP in order to consider spatial and diurnal variability in retrieval biases and investigate the seasonal and diurnal cycles of CTT across the entire SEVIRI disc. This is the first study to relate the diurnal variations in retrieval bias to observed diurnal cycles in CTT for SEVIRI cloud top properties as contained in the 12-year spanning SEVIRI dataset. The SEVIRI dataset is introduced in Sect. 2. Based on this dataset, spatial and seasonal patterns in CTT are examined in Sect. 3.1. In Sect. 3.2 the SEVIRI cloud top temperature data is compared to

CALIOP measurements, extending on existing validation analyses in order to consider the implications of spatial and diurnal variations in retrieval bias for the SEVIRI-based diurnal cycles of CTT. In Sect. 3.3, the diurnal variability of cloud top temperature is quantified across the SEVIRI disc. Conclusions are presented in Sect. 4.

## 2   Data

The analysis presented here uses data from the SEVIRI instrument on the geostationary MSG satellite. SEVIRI is an imager

centred at approximately 0° longitude, with 12 spectral channels in the visible, near-infrared and infrared. The instrument has a temporal resolution of 15 minutes (5 minutes in rapid scan mode covering a limited area) and a spatial resolution ranging from 3 km at satellite nadir (1 km in its high-resolution visible channel) to 11 km at the edge of its field of view.

    The European Organisation for the Exploitation of Meteorological Satellites (EUMETSAT)'s Satellite Application Facility on Climate Monitoring (CMSAF) has produced an updated twelve year dataset of cloud top properties based on SEVIRI

measurements, named the CLoud property dAtAset using SEVIRI, Edition 2 (CLAAS-2) (Benas et al., 2016). CLAAS-2 contains the only retrieval of cloud top properties currently available at full SEVIRI spatial and temporal resolution and over a period of several years. The specific dataset used in this study is the instantaneous cloud top parameters product (CTX version 002) (Benas et al., 2016). The dataset is available for the period 2004-2015 and retrieved at full SEVIRI spatial and temporal resolution.

The retrieval algorithm applied to produce the CLAAS-2 dataset was developed in the framework of the EUMETSAT Satellite Application Facility on Nowcasting (NWCSAF). The full algorithm (NWCSAF/MSGv2012) is documented in Derrien (2013). To summarize, a multi-spectral threshold method, applying a variety of threshold tests in different channels, is used to obtain a pixel-resolution cloud mask. These tests vary according to conditions such as solar illumination, satellite angle and surface type. Cloud type and cloud top properties are also retrieved for pixels classed as fully cloudy or cloud-contaminated.

No further retrievals are made for clear sky pixels, or pixels classified as containing broken clouds.

    The NWCSAF/MSG retrieval calculates the vertical placement of clouds for fully cloudy pixels via a CTP retrieval, which varies according to cloud type, atmospheric conditions, and the data available.



In the case of optically thick clouds, the vertical position of the cloud top can be calculated from measurements of infrared brightness temperatures in atmospheric window channels by simply correcting for above cloud atmospheric absorption (Smith and Platt, 1978). However, in the case of partially cloudy pixels, or semi-transparent cloud, surface radiation may be transmitted through the cloud, or gaps in the cloud cover. In such cases a multi-spectral approach is needed (Smith and Platt, 1978; Menzel et al., 1983; Schmetz et al., 1993).

For opaque clouds at all heights, CTP is therefore diagnosed by comparing observed 10.8 μm brightness temperatures to values simulated by the Radiative Transfer Model for TOVS (RTTOV) radiative transfer model. For high, semi-transparent cloud the infrared window intercept (Schmetz et al., 1993), or radiance rationing (Menzel et al., 1983) retrieval methods are attempted. Full details of the retrieval algorithm in other cases can be found in Derrien (2013).

Finally, for all cloud types, cloud top temperature and height are calculated from cloud top pressure using input from ERA-Interim (ECMWF Reanalysis) reanalysis fields.

Benas et al. (2016) compare the CLAAS-2 CTT data to measurements from the CALIOP instrument between 2006 and 2015. The comparison is made for the CALIOP cloud layer at which the vertically integrated cloud optical depth (COD) is at least 0.2. For this setting they find a mean bias of 2.1 K and a bias corrected root mean squared error (RMSE) of 16.3 K. Bias and RMSE amount to 11.4 K and 22.1 K when no COD thresholds are applied.

In this study we use CLAAS-2 monthly mean diurnal cycle (MMDC) CTT products, provided at a spatial resolution of 0.25° and a temporal resolution of one hour, to quantify the diurnal cycle of CTT across the SEVIRI disc. We also validate instantaneous CTT retrievals (as produced by the NWCSAF/MSG algorithm and included in the CLAAS-2 dataset) against CALIOP, in order to investigate the implications of both the spatial and diurnal variability in the retrieval bias for the accurate quantification of diurnal cycles in cloud top temperature.

All SEVIRI data used in this study are drawn from the CLAAS-2 cloud top temperature dataset, retrieved from SEVIRI observations using the NMCSAF/MSGv2012 algorithm. For clarity, the term 'SEVIRI' will be used to refer to this dataset hereafter.

## 3 Results

### 3.1 Mean cloud top temperature

Seasonal mean SEVIRI cloud top temperatures for the period 2005-2015 are shown in (Fig. 1). As expected, the warmest CTTs are observed over the ocean and the coldest over land, where a strong diurnal cycle in land surface temperatures drives convective initiation. Typical cloud regime patterns, showing deep convection over land in the region of the intertropical convergence zone (ITCZ), shallower convection over the central Atlantic ocean in the trade wind convergence zone and stratocumulus cloud in the southeast Atlantic ocean are evident.

Seasonal patterns in convection, driven by the movement of the (ITCZ) (Waliser and Gautier, 1993; Yang and Slingo, 2001; Schröder et al., 2009) can be clearly seen. In December, January and February (DJF), the ITCZ is shown as a band of cold cloud running from the southern Indian ocean, through central Africa (where it crosses the equator) and the West





African coast before falling back below the equator, towards South America. In June, July and August (JJA), the ITCZ traces a more northerly position, largely located above the equator, stretching from the Gulf States, through the Sahel and trade wind convergence region, towards Venezuela.

In all seasons, regions with the coldest clouds have seasonal mean CTTs of between 200 K and 240 K, indicative of persistent deep convection in these areas. These clouds are concentrated in central Africa, the Amazon and the West African coast. The warmest CTTs are found in the region of persistent stratocumulus cloud in the southeast Atlantic Ocean, where seasonal mean CTTs range between 270 K and 290 K.

The position of the central Atlantic trade wind convergence zone is closely related to the seasonal movement of the ITCZ. Cloud top temperatures in this region fall to between 230 K and 250 K, with particularly cold clouds observed in March, April and May (MAM) and September, October and November (SON) due to the passage of the ITCZ. This indicates the presence of shallower convective cloud, initiated by the convergence of northern and southern hemisphere winds.

The concentration of cold clouds in Central and West Africa suggests that the absolute diurnal cycle of convection (the mean change in CTT throughout the day) is likely to be strongest in these regions, due to the strong vertical development of deep convective clouds. However, before attempting to quantify the diurnal cycle of CTT at cloud top using SEVIRI CTT data, it is necessary to consider the impact of spatial and diurnal variations in retrieval biases, which may have a significant impact on the diurnal cycle derived from this dataset.

### 3.2 Evaluation of SEVIRI cloud top temperature retrievals with CALIOP data

Biases in the CTT retrieval can be expected to display significant temporal and spatial variation. For example, Fig. 1 shows clear spatial and seasonal patterns in cloud type, while surface emissions of longwave radiation also display spatial and temporal variations, particularly over land (Harrison et al., 1990; Wild et al., 2014). The implications of cloud type and land surface emissivity for the accuracy of cloud top property retrievals from SEVIRI were discussed in Sect. 2.

This study considers the implications of both the spatial and diurnal variability in the retrieval bias for the accuracy of diurnal cycle measurements across the SEVIRI disc. To this end one year of SEVIRI and CALIOP CTT retrievals were compared across the SEVIRI disc. This analysis was carried out using data from 2007, the first full year for which CALIOP data are available. Data was processed for a single year of the twelve year CLAAS-2 dataset, balancing the need to process sufficient data points to be able to examine the spatial variability of retrieval bias with the large computational expense of collocating two large datasets.

### 3.2.1 Collocation

Although CALIOP is in sun-synchronous orbit, gathering data only at 13:30 and 01:30 LST (Winker et al., 2006), it provides a very accurate measurement of CTH, and hence CTT, due to its active measurements. It is therefore an excellent dataset for assessing the accuracy of SEVIRI cloud top retrievals. CALIOP also offers the advantage of a long-running dataset (2006 - present) and global coverage, allowing data to be compared across the entire SEVIRI disc over a long period of time.





CALIOP has a vertical resolution of between 30 and 60 m depending on the cloud's height in the atmosphere and is capable of detecting cloud layers with optical depths of 0.01 (McGill et al., 2007; Vaughan et al., 2009). It measures CTH and subsequently uses the GEOS-5 (Goddard Earth Observing System Model, Version 5) atmospheric GCM model to convert form CTH to CTT (NASA, 2013). This conversion can be seen as a potential source of uncertainty in the CALIOP CTT representation, since the model has a coarser vertical and horizontal resolution than CALIOP. Additionally, rising air parcels such as those found in convective clouds) are usually warmer than the surrounding air, as represented by the grid mean temperature of the model fields. The CALIOP dataset used in this study is the same product used in the CLAAS-2 validation report Benas et al. (2016), the Lidar, Level 2, 5 km Cloud Layer, Validated Stage 1 Version 3 product (CAL LID L2 05kmCLay-ValStage1-V3-01) (NASA, 2013).

As SEVIRI's detection efficiency decreases at low optical depths, it is necessary to exclude very thin cloud layers from this comparison. Previous comparisons of SEVIRI and CALIOP data have excluded all cloud with an optical depth of less than 0.3 (Kniffka et al., 2013), 0.2, (Benas et al., 2016) and 0.1 (Stubenrauch et al., 2010; SAFNWC/MSG, 2012), while others have not excluded thin cloud at all (Reuter et al., 2009). In this analysis, mean statistics were calculated for a number of different collocation criteria. Due to the computational expense of collocating the datasets, different collocation criteria were tested using data for every 10th day in 2007.

Table 1 contains information on the seven different sets of collocation criteria tested. It indicates the maximum time window during which retrievals could be collocated, whether multi-layer clouds were included in the comparison and what COD threshold was used. Each set of criteria is identified by an abbreviation, which we use to refer to individual scenarios in the text and by a symbol, which we use to refer to scenarios in the figures.

The mean bias and RMSE for each of the collocation criteria in Table 1 are shown in Fig. 2. Statistics are plotted separately for all collocated data points, land retrievals and ocean retrievals. The number of SEVIRI CTT retrievals collocated with CALIOP for each of the sets of criteria and the number of land and ocean retrievals are shown in Fig. 3.

Adjusting the maximum time window for collocation from 60 to 15 minutes (60-ML-0 and 15-ML-0) does not have a large effect on the mean bias and RMSE (Fig. 2), or on the spatial distribution of the biases (Appendix A). However, Fig. 3 shows that a 15 minute collocation window reduces the number of collocated retrievals by 50%. A temporal collocation window of 60 minutes was therefore chosen for this analysis.

The effects of applying different COD thresholds (60-ML-0, 60-ML-03, 60-ML-1, 60-ML-2) (in order to account for differences in the sensitivity of the SEVIRI and CALIOP instruments to low cloud optical depths) were also considered. However, these thresholds do not affect biases between the radiometric cloud top observed by SEVIRI and the physical cloud top observed by CALIOP. Thresholds were applied to the COD of the top cloud layer as measured by CALIOP and scenes for which this top layer did not meet the threshold value were excluded from the analysis. This differs from the approach implemented by Benas et al. (2016), although it does not lead to a large difference in the spatial distribution of the mean biases (Appendix B).





The mean bias for combined land and ocean data is reduced from 11 K when no COD threshold is used to 2.4 K when a threshold of 0.3 is used (Fig. 2). This is further reduced to 0.4 K for a threshold of 1 and -0.9 for a threshold of 2. Similar decreases in the RMSE were observed (Fig. 2).

The total mean and ocean-only biases are negative for the 60-ML-2 collocation criteria, while the land values display a slight positive bias (Fig. 2). This is due to the fact that the majority of the clouds in this dataset are located in the southeast Atlantic Ocean (Fig. 4), a region of prevalent subtropical subsidence inversions. In the presence of a low level thermal inversion there are two possible solutions when using observed brightness temperatures to infer the vertical placement of cloud tops. As explained in Sect. 2, the SEVIRI retrieval algorithm (NWCSAF/MSGv2012) places the cloud top for low clouds at the pressure level which corresponds to the best fit between observed and simulated brightness temperatures. In cases of low level thermal inversions, the SEVIRI retrieval generally places the cloud above the thermal inversion (Derrien, 2013) and may therefore underestimate CTT in these areas. However, the CALIOP retrieval of CTT is based on direct observations of CTH and is therefore not subject uncertainty with respect to the vertical placement of the cloud top in the first place (NASA, 2013; Hamann et al., 2014). However, the vertical resolution of the model fields used to convert CALIOP CTH to CTT is relatively coarse, which may introduce some uncertainty in the CALIOP values themselves. If the latter effect is small, the difference in approach to the vertical placement of clouds in the presence of inversions would result in a systematic negative bias in the SEVIRI retrieval in this region.

Finally, the impact of excluding multi-layer cloud from the collocation is considered (60-ML-0, 60-SL-0, 60-ML-1 and 60-SL-1, Table 1). Excluding multi-layer cloud when no COD is used (60-SL-0) results in the mean bias falling from 11.1 K to 3.9 K and the mean RMSE falling from 26.3 K to 20.6 K (Fig. 2), indicating that in the case of multi-layer cloud scenes, observed cloud top brightness temperatures are likely contaminated by longwave emissions from lower-level clouds. However, once a COD threshold of 1.0 is applied, excluding multi-layer cloud (60-SL-1) results in a much smaller change in the bias from 0.4 K to -0.4 K, with the mean RMSE falling from 11.8 K to 10.9 K (Fig. 2). This shows that, when the top cloud layer observed by CALIOP has a COD greater than one, the brightness temperatures observed by SEVIRI are no longer significantly contaminated by longwave emissions from lower-level clouds. CALIOP scenes containing multiple layers of cloud were therefore included.

To summarize, for all further analysis presented in this paper, we collocated SEVIRI and CALIOP data using the 60-ML-1 criteria, consisting of a 60 minute collocation window, inclusion of scenes with multiple layers of cloud, and a COD threshold of 1.0. The resulting cloud top temperature retrievals from SEVIRI and CALIOP were compared for the full year of 2007, as follows. For each 5 by 5 km CALIOP pixel within the SEVIRI disc:

- the highest reported CALIOP cloud layer was selected;

- the SEVIRI pixel with the closest latitude and longitude to that of the CALIOP pixel was selected;

- for this pixel, the nearest SEVIRI retrieval in time (within the allowed 60 minute time window) was identified;

- if multiple CALIOP retrievals fell within a single SEVIRI pixel, the values were averaged. This was only necessary towards the edges of the disc, beyond approximately 50°E and 50°W;




– the retrieval bias (calculated as SEVIRI minus CALIOP CTT) was calculated for each instance of collocated data.

### 3.2.2 Retrieval biases

The annual mean bias in SEVIRI CTT, calculated across the entire SEVIRI disc from January to December 2007, combining both daytime and nighttime conditions, is 0.44 K with a standard deviation of 11.7 K (Table 2). This is smaller than the 2.1 K
mean bias calculated by Benas et al. (2016). The difference is likely due to the fact that a less stringent COD threshold of 0.3 was applied by Benas et al. (2016).

Statistics calculated over a variety of regions and conditions show small mean biases and large standard deviations. Over the ocean the mean bias in SEVIRI CTT is smaller, at -0.12 K, with a standard deviation of 10.5 K, but over land the mean bias rises to 2.38 K, with a standard deviation of 14.9 K (Table 2).
It should also be noted that the sign of the mean bias is negative over the ocean for both daytime and nighttime retrievals, although the bias is always positive over land. As discussed in Sect. 2, even once clouds with a low optical depth are filtered out, observations from SEVIRI are expected to detect a warmer radiometric CTT as compared to the colder physical cloud top temperature observed by CALIOP.

The spatial distribution of the number of fully cloudy pixels for which SEVIRI retrieved a CTT value and for which a
corresponding CALIOP value was available during 2007 are shown in Fig. 4. Collocated retrievals are concentrated in the southeast Atlantic Ocean (Fig. 4), an area of almost perpetual cloudiness, where atmospheric inversions are prevalent. The negative biases over the ocean are again due to the different retrieval processes for SEVIRI and CALIOP with regard to the vertical placement of clouds in regions of atmospheric inversions (as discussed in Sect. 3.2.1).

Figure 5 shows the spatial distribution of mean SEVIRI minus CALIOP CTT for daytime, nighttime, day and nighttime
combined, and the difference between night and daytime biases. It can immediately be seen that for both day and night time retrievals, there are large areas of very high mean bias, although, due to compensating biases, these are obscured in some areas when day and nighttime biases are plotted together. SEVIRI is shown to generally overestimate CTT over land, by approximately 10-20 K during the day and 15-25 K (or more) at night (Fig. 5). These values are larger than those reported by Benas et al. (2016) who do not consider day and nighttime biases separately. However, these biases are in agreement with
the expected discrepancy of 0.5-3.0 km (approximately 3-20 K assuming a 6.5 K/km lapse rate) between radiometric and physical cloud top height found by Sherwood et al. (2004); Minnis et al. (2008); Stubenrauch et al. (2010, 2013) in the case of high clouds.

Over the ocean, biases are relatively small. CTT is underestimated by 5-10 K over large areas of the Atlantic ocean, while in other areas, such as the region of trade wind convergence, SEVIRI overestimates CTT by 5-15 K (Fig. 5).
The large areas of slightly negative bias in the Atlantic ocean correspond to the areas of persistent atmospheric inversion. As explained previously, the small systematic bias in the SEVIRI CTT retrieval in this region is due to its treatment of subsidence inversions. As can be seen from Fig. 4, the majority of the collocated retrievals are located in this region. Therefore, the mean biases presented in Table 2 are heavily weighted towards the small negative biases in this region.





The difference in the magnitude of the biases over land and ocean is likely due to differences in the most common cloud regimes observed over land and ocean (Yang and Slingo, 2001; Schröder et al., 2009), as well as the greater difficulty in accounting for variations in surface emissivity over land (Derrien, 2013). As suggested by Sherwood et al. (2004) and Stubenrauch et al. (2013), the differing extinction profiles and vertical heights of convective and stratiform clouds results in larger
differences between the radiometric and physical cloud top for tall convective clouds.

Unfortunately, it is not possible to fully characterize the diurnal variation in bias, as CALIOP data is only available for comparison at 01:30 and 13:30 LST. However, the two CALIOP overpasses can give at least an estimate of the potential size of this variation. For the purposes of this study, a bias in CTT which remained constant throughout the day would not be a barrier to quantifying either the amplitude of the diurnal cycle in CTT, or the average time of minimum CTT. However, the fact that
biases can change dramatically from the day to nighttime overpasses of CALIOP is more problematic.

Both the mean values and the spatial distribution of the biases change significantly from day to night. Differences between mean nighttime and daytime biases in the SEVIRI CTT retrieval can be as large as 30 K in some areas (Fig. 5). There are strong positive differences over Sub-Saharan Africa and South America, with strong negative differences over the Sahara. Differences between night and daytime biases are generally smaller over the ocean and over Europe. In areas where the difference between
daytime and nighttime biases (Fig. 5) is greater than, or equal to the observed magnitude of the diurnal cycle in cloud top temperature, the diurnal cycle observed from the retrieval may be a product of diurnal variability in the accuracy of the retrieval. It will therefore be necessary to consider the diurnal variability in retrieval bias when quantifying the diurnal cycle of cloud top temperature.

### 3.3  Diurnal cycle of CTT

The CLAAS-2 MMDC product was used to calculate three-month mean diurnal cycles of cloud top temperature across the SEVIRI disc. Data were averaged for the period 2005-2015 to produce a diurnal cycle with a temporal resolution of one hour, on a spatial grid of 0.25°. Ten years of data were required in order to produce relatively smooth diurnal cycles, particularly over land, where cloud retrievals were relatively sparse.

An example of the resulting diurnal cycle in cloud top temperature is shown in Fig. 6 for a grid box centred on 3.1°S, 16.4°E,
(western Democratic Republic of the Congo, see cross in Fig. 10) for the months of SON. The amplitude of the diurnal cycle in CTT was calculated as the maximum minus minimum diurnal mean cloud top temperature, as shown by the arrow indicating an amplitude of 30 K. The local solar time of the minimum cloud top temperature was defined as the time at which the minimum daily mean CTT occurred, as shown by the dashed line at 19:00 local solar time (LST).

While the averaging process produced a coherent diurnal cycle in the majority of cases, the calculated diurnal cycle remained
very noisy in a few areas, particularly during seasons when very few clouds were retrieved. The number of CTT retrievals is greatest over areas of the ocean where large, homogeneous stratiform cloud fields result in a large number of cloud-filled pixels and hence a large number of SEVIRI CTT retrievals, and in the region of the ITCZ, where convective cloud is concentrated (Fig. 7). In areas with very few cloud retrievals, such as the Sahara in all seasons and parts of southern Africa in JJA (Fig. 7), it will not be possible to accurately calculate a diurnal cycle of convection.



Maps of the amplitude of the diurnal cycle in SEVIRI cloud top temperature (Fig. 8), calculated as shown in Fig. 6, show the smallest amplitudes located over the southeast Atlantic ocean in all seasons. Over the course of a typical day, stratocumulus cloud tops vary by less than 5 K in this region. Amplitudes increase to between 20 K and 30 K in the trade wind region, where there are more convective clouds. Over Africa and South America amplitudes generally range between 20 K and 50 K, with the seasonal changes tracking the movement of the ITCZ, seen as a migrating band of cold cloud tops in Fig. 1. The diurnal cycle is smaller in Europe where amplitudes range from 15 K in the north during DJF to 50 K in the Mediterranean during JJA.

For illustrative purposes, amplitudes of the diurnal cycle in CTT were plotted for areas with few CTT retrievals, but regions with retrievals at less than 15% of the processed time steps are indicated (Fig. 8). With the exception of southern Africa in JJA, these regions correspond to areas of desert and calculated amplitudes tend to be very large. Amplitudes exceed 60 K in areas of the Sahara and Namibian deserts in all seasons, as well as in Somalia during the December to March dry season (Higgins et al., 1978) and Southern Africa during the May to September dry season (Higgins et al., 1978) (Fig. 8). The large amplitudes observed in these areas are likely to be caused by a mixture of insufficient data (Fig. 7) and, particularly in the Sahara, by a large variation in the size of the retrieval bias between night and daytime conditions (Fig. 5).

In order to consider the effects of systematic differences in day and nighttime CTT retrieval biases in the SEVIRI dataset, the ratio of the amplitude of the diurnal cycle (Fig. 8) to the diurnal variability in the retrieval bias (Fig. 5) was calculated. In regions where this ratio is low, differences between systematic retrieval biases under day and nighttime conditions may contribute strongly to the amplitude of the observed diurnal cycle in cloud top temperatures. A threshold value of 5 was chosen for this ratio, as indicative of regions in which observed diurnal cycles in CTT may simply be artefacts of the diurnal variation in retrieval errors. Areas for which the ratio falls below this threshold are indicated in Fig. 8.

Maps of the phase of the diurnal cycle, defined at each grid box as the local solar time at which the minimum three-monthly mean CTT occurs (Fig. 6) are shown in Fig. 9. Regions with few clouds, or where diurnal variability in retrieval bias may significantly contribute to the observed diurnal cycle are illustrated in Fig. 9, as described for Fig. 8.

Over large areas of the ocean, minimum CTTs are observed at around 16:00. In the southeast Atlantic however the minimum is observed in the morning, at around 09:00.

In South America, minimum CTT generally occurs at around 20:00. Over Sub-Saharan Africa and Europe the minimum is generally observed at around 16:00, with some areas, particularly West Africa, the Sahel and parts of the Congo Basin, showing later peaks at around 18:00. These later peaks broadly track the movement of the ITCZ (Fig. 1) and could be due to more vigorous convection, persisting until later in the day. It could also be due to a mixture of different convective cloud types, including organised MCSs which can persist until the early morning and more isolated local convective cells which peak in the afternoon (Rickenbach et al., 2009; Pfeifroth et al., 2016).

Areas with few cloud retrievals, such as the Sahara desert in all seasons, and southern Africa in JJA, are noisier (Fig. 9). This may be due to the fact that there is simply not enough data in these regions to meaningfully diagnose the phase of the diurnal cycle in CTT. However, the regions with the fewest retrievals (Fig. 4 and Fig. 9) do not match exactly the regions of noise in Fig. 9. This indicates that the noise may also be caused by a mixture of different cloud types with different diurnal cycles.





The relationship between the observed amplitude and phase of the diurnal cycle and the retrieval biases presented in Sect. 3.2 were examined in more detail over the Sahara, central Africa and southeast Atlantic ocean. These areas were chosen because they all exhibit fairly consistent patterns of both retrieval bias and observed diurnal cycle properties and were designed to cover approximately $9 \times 10^6$ km$^2$ each. They also provide examples of desert, rainforest and ocean surface types. The locations of
these three areas are illustrated in Fig. 10.

Seasonal mean SEVIRI diurnal cycles and retrieval biases for each of the regions in Fig. 10 were compared (Fig. 11). We have already shown that the SEVIRI dataset has different retrieval biases under daytime and nighttime conditions (Sect. 3.2), due to differences in solar illumination, cloud types, the availability of visible channel observations and, subsequently the exact retrieval algorithms used. Seasonal mean times of sunrise and sunset are therefore indicated for each region and mean retrieval
biases, as calculated in Sect. 3.2 for the year 2007, are shown for both day and nighttime CALIOP overpasses.

In the southeast Atlantic Ocean, the bias is shown to be very small with no apparent diurnal cycle in the bias (Fig. 11). Mean CTTs reach a minimum at around 09:00 and persist until 16:00 in the DJF and MAM seasons. In JJA the cold clouds are more short-lived and in SON cloud top temperatures remain constant throughout the day.

In the Sahara, the amplitude of the diurnal cycle is almost 20 K, with a small diurnal cycle in the bias of around 5 K (Fig.
11). The warmest CTTs are observed at 05:00 and at noon, with the coldest cloud at 07:00 and 18:00. Although the diurnal cycle in the bias is less than the amplitude of the diurnal cycle, the bias results in cloud top temperatures retrieved during the day being too warm, indicating that a significant fraction of the amplitude observed over the Sahara may be due to differences between the day and nighttime conditions and hence the differences in the retrieval algorithms applied. Sudden changes in mean CTT around the times of sunrise and sunset in the Sahara are also seen in Fig. 11. In all seasons there is a secondary
minimum in cold cloud top temperatures at around 06:00, about an hour before sunrise. This secondary minimum may indicate the presence of MCSs, or simply a change in values due to the change in retrieval algorithm at this point.

In central Africa, the diurnal cycle in the bias is around 7 K and the amplitude of the diurnal cycle is around 15 K throughout the year (Fig. 11). The warmest cloud is observed at 14:00, with the coldest cloud between 20:00 in DJF and 22:00 in JJA. There is a secondary minimum at 09:00 in all seasons except DJF, which may be caused by the presence of MCS, although it
is also possible that this secondary peak is produced by the switch from night to daytime retrieval conditions. In contrast to the Sahara however, the diurnal cycles in retrieval bias create a smaller amplitude of the diurnal cycle in cloud top temperatures than would otherwise be observed. This size of the bias during the early morning may indicate a larger difference between the radiometric cloud top measured by SEVIRI and the physical cloud top measured by CALIOP due to differences in cloud types throughout the day. While the ratio of the diurnal cycle in mean CTT to the difference in retrieval biases is small, it appears
that in this region the difference in the retrieval bias is acting to reduce, rather than increase, the observed diurnal cycle in CTT.

It is interesting to note that the broad shape of the diurnal cycle curves in the Sahara and Central Africa are similar, although the post-sunrise increase in CTT is delayed in Central Africa relative to the Sahara. This could potentially be caused by the lower surface albedo of Central Africa relative to the Sahara, causing the Sahara to heat up more quickly, producing lower, warmer clouds. Any such effect would be delayed by a few hours in Central Africa due to the buffering effect of the large, dark
rainforests in the region.



## 4  Conclusions

In this study we evaluated SEVIRI cloud top temperature data, as retrieved by the NWCSAF/MSGv2012 algorithm and included in the updated CLAAS-2 dataset, against CALIOP and attempted to quantify spatial and diurnal variabilities in retrieval biases. We also quantified the amplitude and phase of the diurnal cycle in cloud top temperatures observed by SEVIRI. Com-

paring our measurements of the diurnal cycle in mean CTT and retrieval bias we show that diurnal variations in the retrieval bias are often of the same order of magnitude as the amplitude of the observed diurnal cycle. Areas in which there was insufficient data to accurately calculate the diurnal cycle in CTT, or in which the observed cycle was an artefact of retrieval biases, were identified.

SEVIRI and CALIOP data were collocated using a 60 minute collocation window and a COD threshold of 1.0. Scenes with

multiple layers of cloud were included. By collocating SEVIRI and CALIOP CTT retrievals for the whole year of 2007, we show that mean errors in the SEVIRI retrieval can vary from less than 5 K to more than 30 K across the SEVIRI disc, and by up to 30 K between the daytime and nighttime overpasses of CALIOP. However, mean errors across the SEVIRI disc are small, at approximately 0.44 K with a standard deviation of 11.7 K. This shows the importance of considering spatial and diurnal variations in retrieval error when using this dataset.

We believe that the difference between the radiometric cloud tops observed by SEVIRI and the physical cloud tops observed by CALIOP may account for a significant fraction of the biases found in this analysis. As explained in Sect. 1, previous studies indicate that biases of less than 0.5-3.0 km (approximately 3-20 K) could potentially be explained by this difference, even for optically thick clouds. As cloud layers with an optical depth of less than one were not included in the comparison of SEVIRI and CALIOP CTT data, we expect biases to be largest in the case of optically thick clouds with poorly defined edges, such as

glaciated clouds. In addition, the small negative bias observed over the southeast Atlantic Ocean is likely related to uncertainties introduced by the use of models to estimate the vertical placement of clouds for both SEVIRI and CALIOP datasets. Additional biases described in this paper may be due to retrieval errors included in the SEVIRI dataset.

Keeping these uncertainties in mind, the seasonal, diurnal and spatial variation of cloud top temperatures were quantified across the SEVIRI disc. By plotting the seasonal mean amplitude and phase of the diurnal cycle in cloud top temperature,

we show that SEVIRI is able to capture details of the diurnal cycle of convection, across several continents. We show that the CLAAS-2 dataset measures the diurnal cycle of cloud tops accurately in regions of stratiform cloud such as the southeast Atlantic and Europe, where retrieval biases are small and exhibit limited spatial and temporal variability. Quantifying the diurnal cycle over the tropics and regions of desert is more difficult, as biases are larger and more variable.

Looking at three areas in detail (the southeast Atlantic Ocean, the Sahara desert and Central Africa), we analyse the rela-

tionships between the diurnal cycle in cloud top temperature and retrieval biases. We show that retrieval biases in the southeast Atlantic are small enough to detect a small but persistent diurnal cycle of approximately 5 K in the area, with cold clouds peaking between 11:00 and 15:00 local solar time. However, the CLAAS-2 dataset is shown to be of limited skill in measuring the diurnal cycle over the Sahara, which may be due to generally low cloud cover in desert regions and a possible dominance of optically thin clouds such as cirrus outflow from tropical convection when clouds are present. In the Sahara, variability in





the bias appears to contribute to an excessively large amplitude of the diurnal cycle, with a large amount of spatial and seasonal variability in the phase. In tropical regions such as Central Africa, a relatively large variability in the retrieval biases appears to dampen the signal from a very strong observed diurnal cycle, with minimum cloud top temperatures occurring consistently between 20:00 and 22:00.

While this study highlights the importance of considering spatial and diurnal variations in retrieval errors when using SEVIRI data, it is also the case that observations from passive imagers in geostationary orbit provide valuable observations of the temporal and spatial variability of cloud on scales which are not available from polar-orbiting satellites such as CALIOP. We therefore see our results as guidance for the observation and modelling communities when using SEVIRI cloud top properties, particularly for studies considering the diurnal cycle of cloud top properties.

**Appendix A:  Comparing results of different collocation time windows**

The insensitivity of the calculated bias in SEVIRI CTT to a change in the collocation window used from 60 minutes (i.e. a maximum temporal distance between SEVIRI and CALIOP retrievals of 30 minutes) to 15 minutes (i.e. a maximum temporal distance between retrievals of 7.5 minutes) is initially surprising. We collocated SEVIRI and CALIOP CTTs, for the full year of 2007, using both 60 minute and a 15 minute collocation windows. This amounts to an extra 22.5 minutes between CALIOP

and SEVIRI retrievals in the 60 minute window case, as compared to the 15 minute case.

There is no significant change in either the magnitude, or spatial distribution of the observed biases between the two cases (Fig. 5 and Fig. A.1). However, by reducing the collocation window to 15 minutes, the number of collocated data points is reduced and the spatial patterns become less clear.

The biases shown in Figs. 5 and A.1 consist of biases due to differences in the retrieval processes of the SEVIRI and

CALIOP datasets (the retrieval bias) and to spatial and temporal differences in the scenes observed by the two instruments (the collocation bias).

If the size of the retrieval bias increased when moving from a 15 to 60 minute collocation window, we would expect either the mean biases in Fig. 5 to be larger than those shown in Fig. A.1, or, if the mean values are obscuring compensating errors (for example from observations before and after the CALIOP overpass), for the standard deviation of the retrieval biases in the

60 minute case to be larger than those in the 15 minute case. However, there is little difference between the two sets of maps (Fig. A.2). This indicates that the size of the collocation biases does not increase significantly when using a 60 minute time window in place of a 15 minute window.

There are many reasons to think that the collocation bias may be small relative to the large retrieval biases seen in many parts of the SEVIRI disc. For example, over areas of stratiform cloud, cloud top temperature is unlikely to change significantly

over the space of the extra 22.5 minutes allowed by a 60 minute collocation window. In more convective areas, some clouds may develop significantly over the course of the larger time window, but the cloud top temperature of mature convective cloud systems and convective anvils will be more stable over time.



## Appendix B:  Cloud optical depth threshold methodology

When collocating SEVIRI and CALIOP retrievals of CTT, scenes observed by CALIOP were excluded if the top cloud layer had an optical depth of less than 1. This differs from the approach implemented by Benas et al. (2016), who compared CTT values for the first CALIOP layer at which the top-down, vertically integrated cloud optical depth exceeded the threshold value.

5 The number of scenes containing cirrus cloud is therefore reduced in this analysis, compared to that of Benas et al. (2016). This is likely to increase the weighting of statistics presented in Sect. 3.2 towards the southeast Atlantic ocean, where there are few cirrus clouds. However, it does not impact the weighting of statistics elsewhere in the study, where the data is not limited to retrievals which can be collocated to a CALIOP overpass. A comparison of mean SEVIRI day and nighttime retrieval biases (Fig. B.1) with a similar plot in Benas et al. (2016) (Fig. 6-15, row 3, column 3) indicates that this difference in methodology

10 does not lead to a large difference in the spatial distribution of the mean retrieval biases.

*Author contributions.*  S.A.T. and P.S. co-designed the study. S.T. analysed the data and wrote the paper. P.S. and B.A.W. provided suggestions for the methodology, discussed results and commented on the manuscript at all stages. S.F and M.S. developed the CLAAS-2 dataset and provided advice on its use.

*Acknowledgements.*  SEVIRI CLAAS version 2 data were obtained from EUMETSAT's Climate Monitoring Satellite Applications Facility

15 (CMSAF). CALIOP data were obtained from the NASA Langley Research Center Atmospheric Science Data Center. This work was supported funding from the European Research Council under the European Union's Seventh Framework Programme (FP7/2007–2013)/ERC grant agreement no. FP7-280025.



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





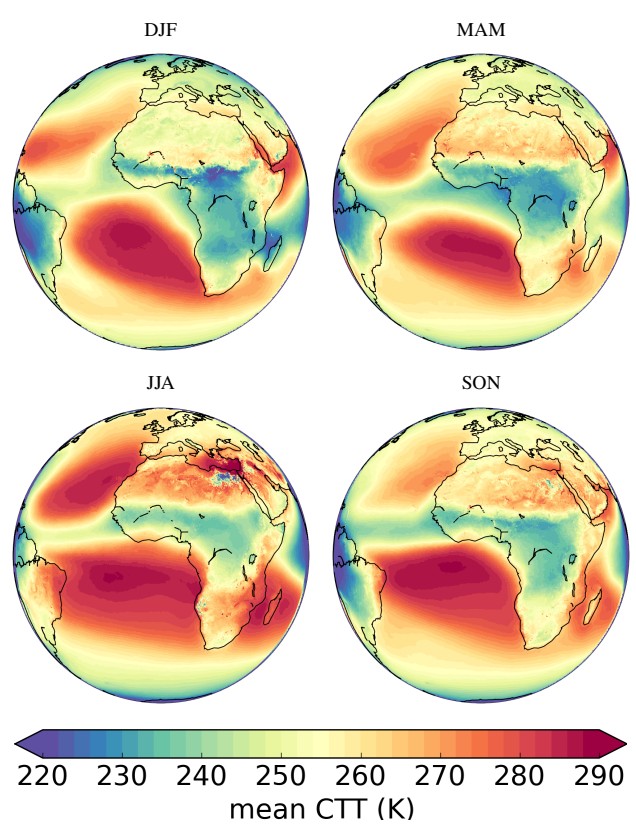

**Figure 1.** Seasonal mean SEVIRI cloud top temperatures for the period 2005-2015.





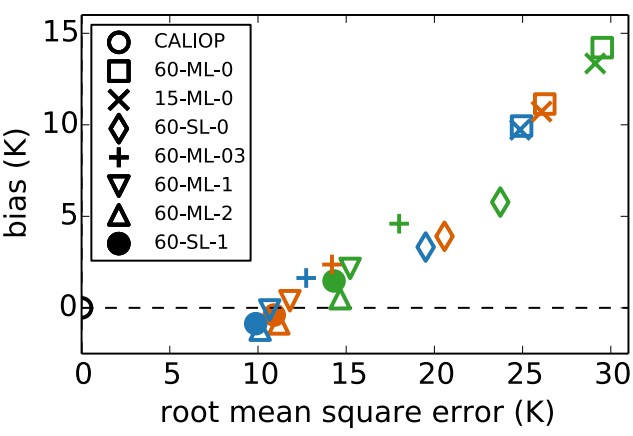

**Figure 2.** Bias (SEVIRI minus CALIOP CTT ) versus root mean square error of SEVIRI cloud top temperature retrievals. Symbols refer to the different sets of collocation criteria, defined in Table 1. Green symbols show retrievals over land, blue over ocean and orange over both. The 'CALIOP' point on the left-hand side indicates where a retrieval which perfectly reproduced the CALIOP observations would be located.





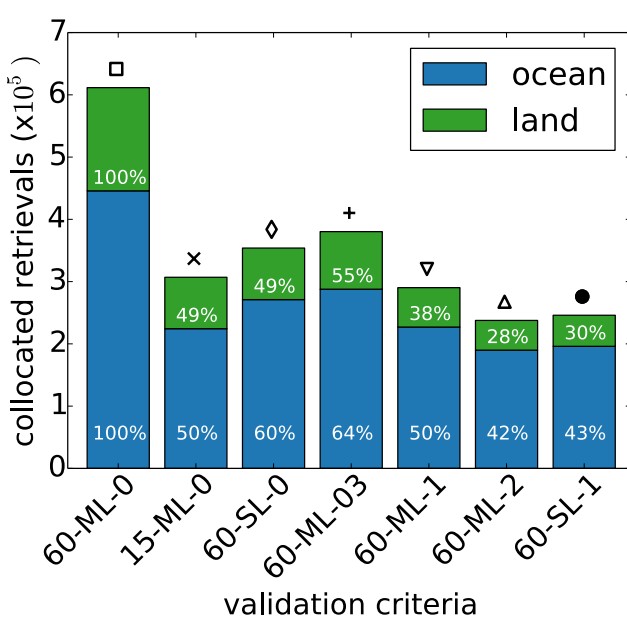

**Figure 3.** Number of SEVIRI and CALIOP retrievals collocated in 2007 for each of the collocation criteria defined in Table 1. Collocation criteria are identified by both text abbreviation and symbol. Colours show the division between land (green) and ocean (blue) retrievals. Percentages show what fraction of the total number of available retrievals are processed for each set of collocation criteria.



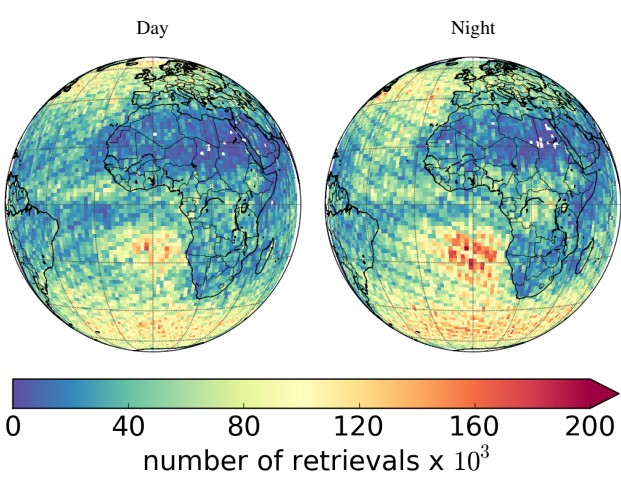

**Figure 4.** Spatial distribution of the number of collocated SEVIRI and CALIOP retrievals in 2007, shown separately for daytime and nighttime conditions.





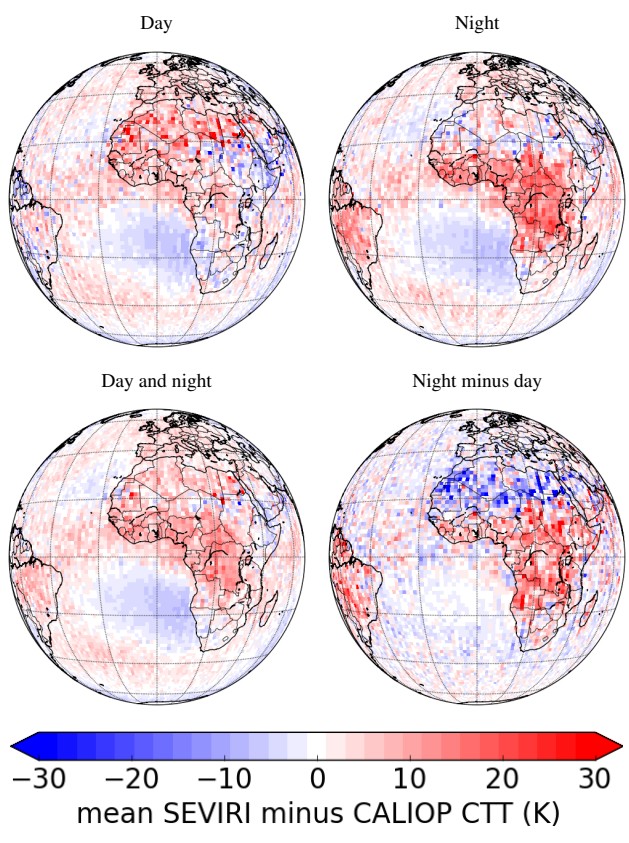

**Figure 5.** Spatial distribution of the bias in SEVIRI cloud top temperature retrievals during 2007. Biases are shown for the day (13:30 LST CALIOP overpass), night (01:30 LST CALIOP overpass), mean of both day and night biases, and for the difference (night minus day) between night and daytime biases.





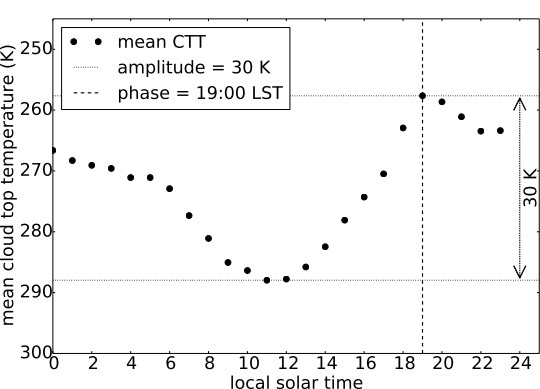

**Figure 6.** Seasonal mean climatological (September, October, November, 2005 - 20015) diurnal cycle of cloud top temperature at 3.1°S, 16.4°E, (western Democratic Republic of the Congo, see cross in Fig. 10). The amplitude of the diurnal cycle (defined as minimum minus maximum CTT) and the phase (defined as the local solar time (LST) of minimum CTT) are also illustrated.





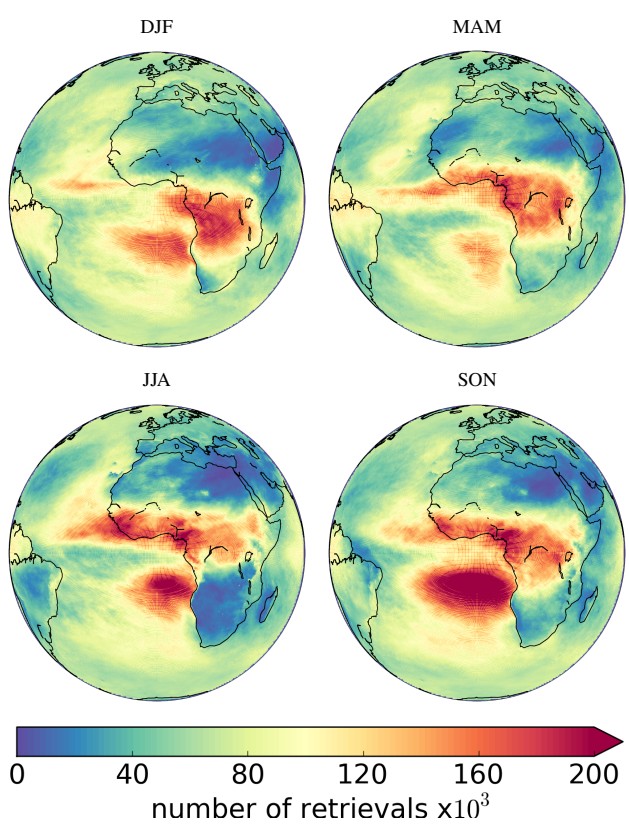

**Figure 7.** Spatial distributions of the total number of SEVIRI cloud top temperature retrievals available from the CLAAS-2 dataset during the period 2005-2015. Values are shown for each season.







**Figure 8.** Seasonal mean amplitude of the diurnal cycle in SEVIRI cloud top temperature for the period 2005-2015. The white overlay indicates regions where retrievals are available for fewer than 15% of the processed timesteps. Squares indicate regions where the ratio of the amplitude of the diurnal cycle in CTT to the diurnal variability in CTT retrieval bias is less than 5.



**Figure 9.** Seasonal mean phase (local solar time of minimum climatological mean CTT) of the diurnal cycle in cloud top temperature for the period 2005-2015. The white overlay indicates regions where retrievals are available for fewer than 15% of the processed timesteps. Squares indicate regions where the ratio of the amplitude of the diurnal cycle in CTT to the diurnal variability in CTT retrieval bias is less than 5.





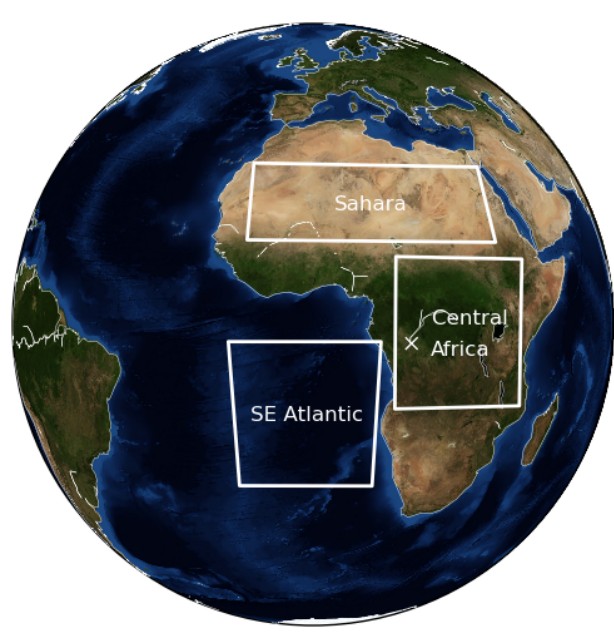

**Figure 10.** Map of the area observed by SEVIRI showing true-colour surfaces, major rivers and lakes. Labelled boxes show the locations of the regions used in Fig. 11. The white cross shows the location of the data used in Fig. 6.



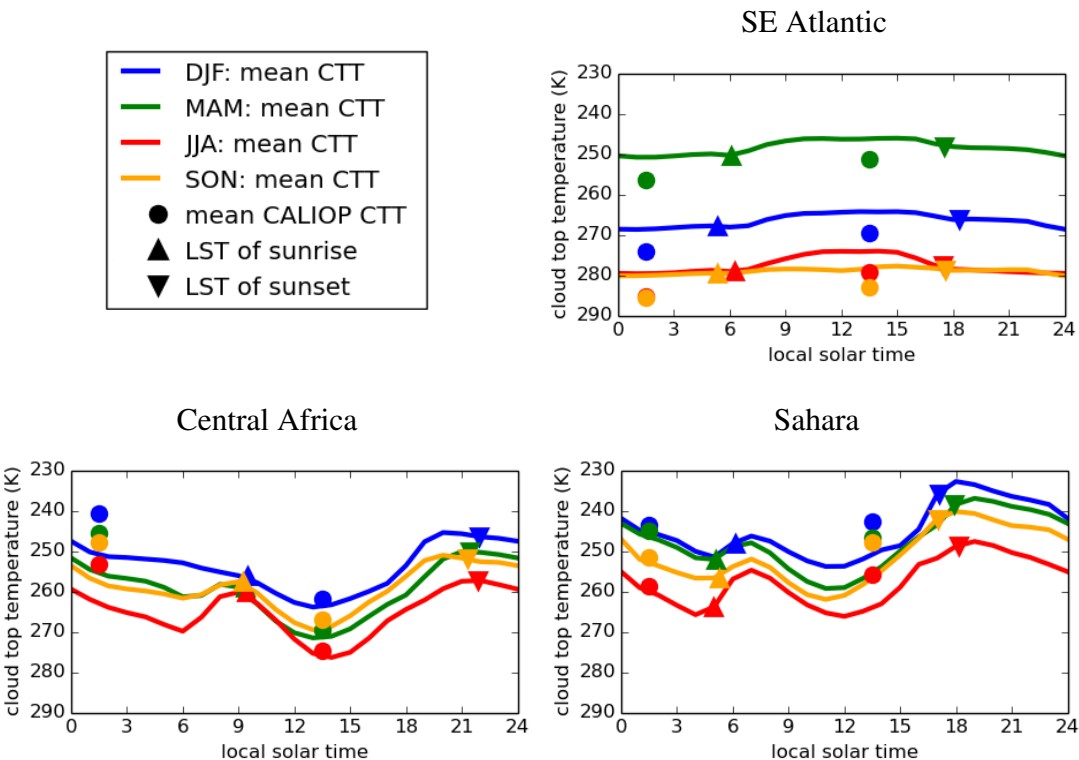

**Figure 11.** Seasonal mean diurnal cycles of SEVIRI CTT calculated over the period 2005-2015 (solid lines) and mean 2007 SEVIRI minus CALIOP retrieval bias (vertical distance between coloured circles and corresponding coloured lines). Biases are shown at the mean LST for the day and nighttime CALIOP overpasses. Also shown are the local solar times of sunrise (triangles) and sunset (nablas) for each region and season.





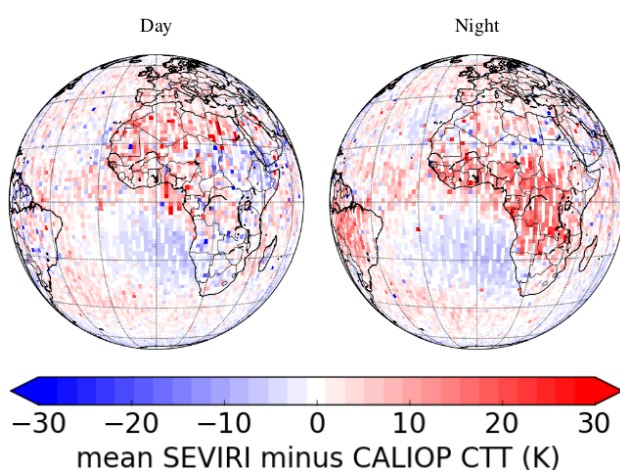

**Figure A.1.** Spatial distribution of the bias in SEVIRI cloud top temperature retrievals during 2007, using a 15 minute collocation window. Biases are shown separately for the day (13:30 LST CALIOP overpass) and night (01:30 LST CALIOP overpass).





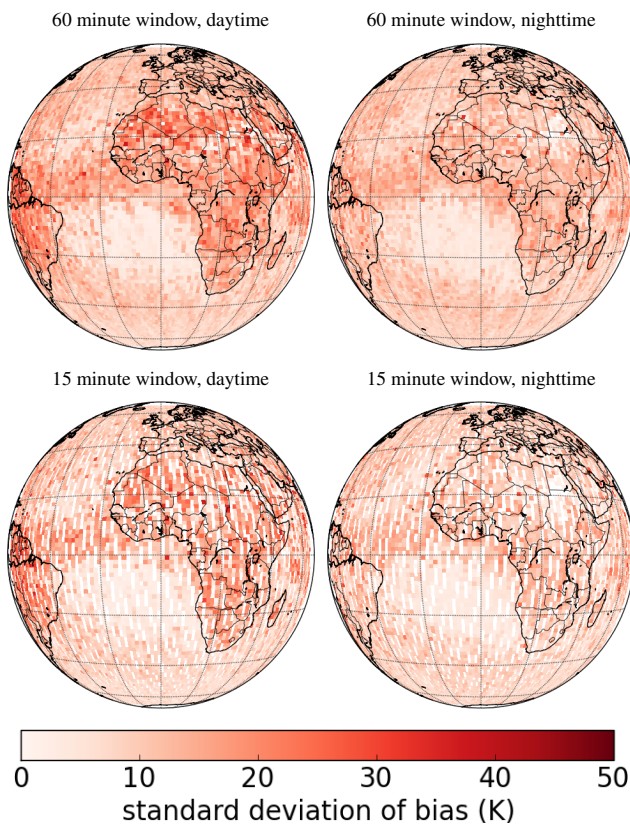

**Figure A.2.** Spatial distribution of the standard deviation of the biases in SEVIRI cloud top temperature retrievals during 2007. Data are shown for both the day (13:30 LST CALIOP overpass) and nighttime (01:30 LST CALIOP overpass), using a 60 minute collocation window and a 15 minute collocation window.





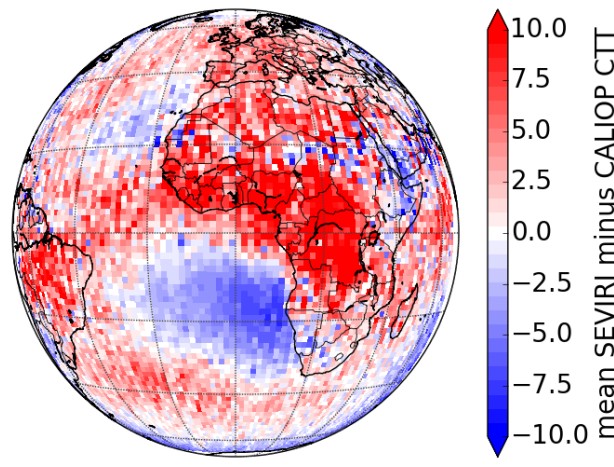

**Figure B.1.** Spatial distribution of mean daytime (13:30 LST CALIOP overpass) and nighttime (01:30 LST CALIOP overpass) bias in SEVIRI cloud top temperature retrievals during 2007. This plot is the same as the 'Day and night' plot in Fig. 5, but plotted using the same scale as Fig. 6-15 in Benas et al. (2016) for ease of comparison.





**Table 1.** Descriptions of the seven sets of collocation criteria to be evaluated, the abbreviations by which they are referenced in the text and the symbols by which they are referenced in plots.

| Abbreviation | Symbol | Collocation window (mins) | Layers included | COD threshold |
|---|---|---|---|---|
| 60-ML-0 | □ | 60 | multi | none |
| 15-ML-0 | × | 15 | multi | none |
| 60-SL-0 | ◇ | 60 | single | none |
| 60-ML-03 | + | 60 | multi | > 0.3 |
| 60-ML-1 | ▽ | 60 | multi | > 1.0 |
| 60-ML-2 | △ | 60 | multi | > 2.0 |
| 60-SL-1 | ● | 60 | single | > 1.0 |





**Table 2.** Summary of evaluation results for the 60-ML-1 collocation criteria, showing: number of collocated SEVIRI retrievals, mean bias (SEVIRI minus CALIOP CTT) and standard deviation of the bias. Statistics are reported separately for day, night, land and ocean retrievals.

| Surface type | Time of day | Number of retrievals ($10^6$) | Mean bias (K) | Standard deviation of bias (K) |
|---|---|---|---|---|
| land and ocean | day and night | 2.79 | 0.44 | 11.7 |
|  | day | 1.34 | 0.05 | 12.2 |
|  | night | 1.45 | 0.80 | 11.2 |
| land | day and night | 0.63 | 2.38 | 14.9 |
|  | day | 0.32 | 0.72 | 16.0 |
|  | night | 0.31 | 4.11 | 13.5 |
| ocean | day and night | 2.16 | -0.12 | 10.5 |
|  | day | 1.02 | -0.16 | 10.7 |
|  | night | 1.14 | -0.10 | 10.4 |