# Peer review of "Evaluating the diurnal cycle in cloud top temperature from SEVIRI"

_Atmospheric Chemistry and Physics, 2016_

## Referee Comment (RC1) · Anonymous Referee #1 · 14 Dec 2016

Review of "Evaluating the diurnal cycle in cloud top temperature from SEVIRI" by Sarah Taylor et al.

Using the CLAAS-2 dataset, the authors characterize the diurnal cycle of cloud top temperature for several regions within the SEVERI's observation disk. Retrievals from SEVERI are compared to CTT inferred from collocated CALIPSO observations in terms of bias and variability. As cloud top temperature is a frequently used quantity in radiative balance and cloud microphysics studies, I feel this is a useful contribution to the literature. I recommend that this study be published after the following minor comments are addressed.

Specific Comments:

P3, L17: "... images such as SEVERI observe the radiometric height of the cloud."

Perhaps it would be helpful to briefly expand upon what is meant by "radiometric height" in the context of a weighting function.

P9, L12: CALIOP doesn't observe the CTT, rather it is inferred from CTH measured by the instrument. Please phrase this differently.

P12, L32-34: This statement does not make sense to me. Why would a region with a lower surface albedo heat up more slowly? Could this be a consequence of a decreased sensible heat flux in Central Africa due to evapotranspiration?

P14, L12-13: While not incorrect, "temporal distance" seems like an unusual way of expressing this, especially since it is primarily a term used in psychology. Perhaps phrasing it as something like "within $\pm 30$ minutes of CALIOP observation" may be clearer.

Technical Corrections:

P2, L4: "cloud" should be plural

P9, L20: "night time" should be "nighttime".

———————————————————

---

## Referee Comment (RC2) · Anonymous Referee #2 · 20 Dec 2016

This study attempts to evaluate retrieval biases in the CLAAS-2 Cloud Top Temperature (CTT) dataset derived from SEVIRI via a comparison with co-located CALIOP retrievals. The novelty of the study involves the separation of the evaluation into daytime and nighttime components, in order to establish whether the quality of the retrievals is consistent through the day. This is important given one of the key benefits of SEVIRI based retrievals should be their ability to capture the diurnal cycle in the geophysical quantity of interest. Having qualified the retrieval quality the authors then investigate the diurnal variability seen within the CLAAS dataset and discuss the implications of their evaluation for the robustness of the CTT signals contained within it.

Overall I think that the concept of the study is a good one – as noted about, the key benefit of CLAAS above what is possible from instruments in polar orbit should be an accurate representation of the diurnal cycle in cloud parameters. Hence making

potential users aware of deficiencies in this representation is useful. The paper is generally well written and the methodology clearly presented.

However, I do have some major and minor comments which I list below. Subject to these being satisfactorily addressed, in my opinion, the paper will be suitable for publication.

Major Comments

1. Nowhere is the accuracy of the 'truth' dataset, CTT from CALIOP, actually quantitatively defined. There is also no mention of whether there is any difference in CALIOP retrieval quality from day to night which I suspect there might be. I also wonder whether the CALIOP inferred heights and optical depths are equally accurate for all cloud types since I would expect the sensitivity to be higher for clouds comprised of smaller ice particles and droplets than for those comprised of larger droplets. Quantitative information concerning accuracy must be in the paper rather than phrases like 'very accurate'.

2. In a similar vein, the transition of height or pressure to CTT using model fields is mentioned as a possible reason for discrepancies between the CALIOP and SEVIRI retrievals, particularly in the SE Atlantic stratus region. However, detail on how the transition is done is rather lacking. It would be useful to know how, for example, timing discrepancies between the background meteorological fields and the satellite overpasses are dealt with. Ideally, in order to isolate the impact of differences in the background model fields one would want to be able to do the analysis using the same set but I appreciate this is beyond the scope of this study. Can any literature be used to give some idea of how well the different meteorological models represent the background state under different regimes as this will affect the CTT comparison?

3. The introduction focuses almost exclusively on convective cloud yet much of the paper discusses regions of stratiform and potentially mixed cloud. Hence the introduction needs some broadening to reflect this.

[Figure]

4. It was unclear to me whether pixels were only considered if they were fully cloud covered or if broken cloud scenes were also considered. I thought the former, but there seemed to be a significant amount of discussion on the effects of surface emission. For the majority of deep convective cases, for fully covered pixels this wouldn't actually have an effect. Since the authors limit the cases studied to greater than an optical depth (where spectrally?) of > 1.0 even outside of those regions the surface impact would be limited.

5. Given much of the manuscript is spent showing deficiencies in the CLAAS retrieval algorithm outputs I feel a short summary of the algorithm is necessary. If this seems too demanding I think at the very least there should be some description of what changes from day to night. I assume that visible channels are used in the daytime (in addition to IR) which may go some way towards explaining why the biases, relative to CALIOP, are improved during the day.

Minor Comments

Page 2 line 32: Consider also adding papers by Pearson et al., 2010 (JGR), 2014 (QJRMS)

Page 3 line 14: It's a little bit of a stretch to say that SEVIRI 'covers' the Middle East as implied here. It certainly sees it but perhaps not all of it. Similar comments apply to the Atlantic Ocean.

Page 3 line 17: Perhaps a little pedantic but SEVIRI does not observe the radiometric height of the cloud, it observes radiances (or, if we want to be completely technically accurate, digital counts). These can then be inverted to estimate the radiometric height. On a similar theme, the manuscript is written as if CALIOP observes cloud top height, which is not technically true, rather it observes backscatter, and, as with any satellite instrument, the required geophysical variable is inferred in some way.

Page 3, line 34: quantify what you mean by optically thin, and give the wavelength that

you are defining the opacity at.

Page 4, line 14: this makes it sound as if you are using data from one instrument flying on one satellite through the whole period considered here which is not true. Neither is it true that throughout this record the operational SEVIRI instrument was at 0 degrees longitude.

Page 4, line 31: more detail on how this CTP retrieval works is required. For example, the wording 'the data available', is extremely vague.

Page 5, line 10: as noted in the major comments, more detail on how this is done is required.

Figure 1: I find this figure, although useful, somewhat misleading as it implies that clouds are ubiquitous. I would suggest another figure or set of panels indicating frequency of cloud occurrence would be beneficial. Similar information is shown later on in the paper but I think it would be good to have it upfront.

Page 6, line 12: while I agree that the diurnal cycle could well be largest here I don't think it necessarily follows from the reasoning given. It could be that the cold clouds are there throughout the day. Figure 1 tells you nothing about this on its own.

Page 6, line 18: Figure 1 implies a spatial and seasonal pattern in cloud type but it doesn't explicitly show it. Since you state that the cloud retrieval you are using gives cloud type it would be interesting to see if that maps to what you see in Figure 1.

Page 6, line 20: I don't think section 2 really gives a quantitative idea of the implications of cloud type and land surface emissivity for the accuracy of CLAAS CTT. There was more of a general discussion in the introduction to be honest.

Page 6, line 25: When the authors say 'data was processed' are they referring simply to the collocation process? It might be good to state this explicitly.

Page 6, line 30: 'very accurate' is not very scientific.

Page 7, line 17: define COD as Cloud Optical Depth (and give wavelength).

Page 7, line 25: I assume the temporal collation window is centred on the CALIOP observation. Is the variation in the timing of SEVIRI scan lines accounted for? What is the spatial match up criterion?

Page 7, line 28: Actually figure 2 suggests that the tightening from 0.3 to 1 does result in a reduced bias while above 1 there is not much change. The authors say this themselves in the next paragraph.

Page 7, line 30: How is a cloud layer defined? i.e. are the thresholds simply applied to the topmost layer with a COD as diagnosed by the CALIOP product? Is there any change in vertical resolution in this product? I'm not sure whether it would have an impact but if for example the vertical resolution reduces with height above a certain point, the same COD would actually indicate a more diffuse extinction profile within the layer.

Page 8, line 13: what is the 'relatively coarse' resolution in numbers? How does this compare to the CALIOP vertical resolution? How well do both of the meteorological models actually capture low level temperature inversions? Do they persist throughout the night and day?

Page 9, line 7: do the deviations show a Gaussian distribution?

Figure 5: I would weight these differences by occurrence or at the very least discuss them in the context of Figure 4. Otherwise the eye is drawn to the very strong positive daytime bias over the Sahara when really there are few clouds there. Why are there negative differences over the east of Africa during the daytime (Sudan/Ethiopia)?

Page 9, line 28: it may be the colour scale but I would say that while the majority of the ocean shows small biases, there are regions where differences look relatively large. One of these regions is discussed in the next paragraph so perhaps merge these two parts together.

Page 10, line 20: I understand that the authors have used three month means in order to obtain sufficient data to see a coherent cycle at the pixel scale but do they perhaps worry that by doing this you are losing information about how the phase of any diurnal cycle in CTT might vary through the year? The discussion about producing 'smooth cycles' is rather vague.

Figure 6: Can the authors provide some idea of the range of values that comprise each hourly mean please, perhaps using quartiles or SDs if the distribution is Gaussian?

Page 10, line 33 (and in other sections focused on the Sahara): I am a bit bemused about the emphasis on the Sahara in the latter part of the paper. As is shown, there is very little cloud being detected there (as one might anticipate) and I am not sure that I would expect too much of what is there to be deep convective in nature (at least north of the inter-tropical front). Hence why should we expect a marked diurnal cycle? Moreover, when what is happening there is analysed the statistics will be poor.

Page 12, line 33: I agree that vegetated Central Africa will typically have a lower surface albedo than the Sahara but I don't see why this would cause the Sahara to heat up more quickly after sunrise (I would actually expect the opposite based on albedo alone) or how this would produce lower, warmer clouds. Please explain.

Page 13, line 22: 'Additional biases...' – this sentence is very vague. What retrieval errors are being referred to?

---

## Author Comment (AC1) · 27 Feb 2017

**Review of "Evaluating the diurnal cycle in cloud top temperature from SEVIRI" by Sarah Taylor et al.**

Using the CLAAS-2 dataset, the authors characterize the diurnal cycle of cloud top temperature for several regions within the SEVERI's observation disk. Retrievals from SEVERI are compared to CTT inferred from collocated CALIPSO observations in terms of bias and variability. As cloud top temperature is a frequently used quantity in radiative balance and cloud microphysics studies, I feel this is a useful contribution to the literature. I recommend that this study be published after the following minor comments are addressed.

> **NB:** Page and line numbers refer to location in the difference file, rather than the revised manuscript.

**Specific Comments:**

P3, L17: "... images such as SEVERI observe the radiometric height of the cloud." Perhaps it would be helpful to briefly expand upon what is meant by "radiometric height" in the context of a weighting function.

> P3, L35: A brief discussion of weighting functions and their relation to radiometric height has been added to the text.

P9, L12: CALIOP doesn't observe the CTT, rather it is inferred from CTH measured by the instrument. Please phrase this differently.

> P11, L27: We have rephrased the text to clarify the method by which CTT are obtained from CALIOP.

P12, L32-34: This statement does not make sense to me. Why would a region with a lower surface albedo heat up more slowly? Could this be a consequence of a decreased sensible heat flux in Central Africa due to evapotranspiration?

> P15, L23: We thank the reviewer for their comment on this point. We revised our arguments and agree with the reviewer that the facts do not support this claim. For this reason, we have dropped this line of argument and leave the investigation of this feature to future studies. We have retained a discussion of the differences in the diurnal cycle between these two regions, and highlighted the lack of a robust explanation for the difference.

P14, L12-13: While not incorrect, "temporal distance" seems like an unusual way of expressing this, especially since it is primarily a term used in psychology. Perhaps phrasing it as something like "within ±30 minutes of CALIOP observation" may be clearer.

> P17, L9: We have amended the text to "within ±30 minutes of a CALIOP overpass", in order to clarify our meaning here.

**Technical Corrections:**

P2, L4: "cloud" should be plural

We have amended the text as suggested.

P9, L20: "night time" should be "nighttime".

We have amended the text as suggested.

---

## Author Comment (AC2) · 27 Feb 2017

**Review of "Evaluating the diurnal cycle in cloud top temperature from SEVIRI" by Sarah Taylor et al.**

This study attempts to evaluate retrieval biases in the CLAAS-2 Cloud Top Temperature (CTT) dataset derived from SEVIRI via a comparison with co-located CALIOP retrievals. The novelty of the study involves the separation of the evaluation into daytime and nighttime components, in order to establish whether the quality of the retrievals is consistent through the day. This is important given one of the key benefits of SEVIRI based retrievals should be their ability to capture the diurnal cycle in the geophysical quantity of interest. Having qualified the retrieval quality the authors then investigate the diurnal variability seen within the CLAAS dataset and discuss the implications of their evaluation for the robustness of the CTT signals contained within it. Overall I think that the concept of the study is a good one – as noted about, the key benefit of CLAAS above what is possible from instruments in polar orbit should be an accurate representation of the diurnal cycle in cloud parameters. Hence making potential users aware of deficiencies in this representation is useful. The paper is generally well written and the methodology clearly presented. However, I do have some major and minor comments which I list below. Subject to these being satisfactorily addressed, in my opinion, the paper will be suitable for publication.

> **NB:** Page and line numbers refer to location in the difference file, rather than the revised manuscript.

**Major Comments:**

1. Nowhere is the accuracy of the 'truth' dataset, CTT from CALIOP, actually quantitatively defined. There is also no mention of whether there is any difference in CALIOP retrieval quality from day to night which I suspect there might be. I also wonder whether the CALIOP inferred heights and optical depths are equally accurate for all cloud types since I would expect the sensitivity to be higher for clouds comprised of smaller ice particles and droplets than for those comprised of larger droplets. Quantitative information concerning accuracy must be in the paper rather than phrases like 'very accurate'.

> P8, L22: We have added information on the accuracy of the CALIOP retrieval of CTH and the process by which this is converted to CTT, including a discussion of the potential inaccuracies introduced by the conversion process. We have also contacted NASA's CALIPSO group and included as much information as is available on the accuracy of the CALIOP cloud layer and CTT retrievals. Unfortunately, no studies have been published on the accuracy of the CALIOP CTT measurements themselves.

2. In a similar vein, the transition of height or pressure to CTT using model fields is mentioned as a possible reason for discrepancies between the CALIOP and SEVIRI retrievals, particularly in the SE Atlantic stratus region. However, detail on how the transition is done is rather lacking. It would be useful to know how, for example, timing discrepancies between the background meteorological fields and the satellite overpasses are dealt with. Ideally, in order to isolate the impact of differences in the background model fields one would want to be able to do the analysis using the same set but I appreciate this is beyond the scope of this study. Can any literature be used to give some idea of how well the different meteorological models represent the background state under different regimes as this will affect the CTT comparison?

> P6, L1: We have extended the description of the SEVIRI algorithms used in Sect. 2, including the use of model reanalysis data. In particular, we have added explanations of how spatial and temporal discrepancies are dealt with and references to additional literature relating to the atmospheric models used. We have also added further information about GEOS-5 model parameters used in the CALIOP cloud top properties retrieval (P9, L2). After discussing this issue with the creators of the SEVIRI and CALIOP datasets, we have not been able to discover any studies on how well these two meteorological models represent the background state under different regimes.

3. The introduction focuses almost exclusively on convective cloud yet much of the paper discusses regions of stratiform and potentially mixed cloud. Hence the introduction needs some broadening to reflect this.

> Sect. 1: We have widened the focus of the introduction to include details of the diurnal cycle of all cloud types.

4. It was unclear to me whether pixels were only considered if they were fully cloud covered or if broken cloud scenes were also considered. I thought the former, but there seemed to be a significant amount of discussion on the effects of surface emission. For the majority of deep convective cases, for fully covered pixels this wouldn't actually have an effect. Since the authors limit the cases studied to greater than an optical depth (where spectrally?) of > 1.0 even outside of those regions the surface impact would be limited.

> SEVIRI pixels were considered only if they were fully cloud covered. We agree with the reviewer that once clouds with an optical depth >1.0 are excluded, surface emissions are unlikely to have a large effect. We have removed unnecessary references to surface emissions.
> The impact of surface emissions is however relevant to the discussion of how to choose a COD threshold when comparing SEVIRI and CALIOP data, as well as to the discussion of previous studies, which have not used such stringent COD thresholds. These references to surface emissions have therefore been retained.

5. Given much of the manuscript is spent showing deficiencies in the CLAAS retrieval algorithm outputs I feel a short summary of the algorithm is necessary. If this seems too demanding I think at the very least there should be some description of what changes from day to night. I assume that visible channels are used in the daytime (in addition to IR) which may go some way towards explaining why the biases, relative to CALIOP, are improved during the day.

> P6, L1: We have extended the description of the algorithms used for cloud masking and cloud top pressure retrievals for the CLAAS-2 dataset in section 2.

**Minor Comments:**

Page 2 line 32: Consider also adding papers by Pearson et al., 2010 (JGR), 2014 (QJRMS)

> P3, L10: We have included the suggested papers in our discussion.

Page 3 line 14: It's a little bit of a stretch to say that SEVIRI 'covers' the Middle East as implied here. It certainly sees it but perhaps not all of it. Similar comments apply to the Atlantic Ocean.

> P3, L31: We have amended the text to clarify that only parts of these regions are covered.

Page 3 line 17: Perhaps a little pedantic but SEVIRI does not observe the radiometric height of the cloud, it observes radiances (or, if we want to be completely technically accurate, digital counts). These can then be inverted to estimate the radiometric height. On a similar theme, the manuscript is written as if CALIOP observes cloud top height, which is not technically true, rather it observes backscatter, and, as with any satellite instrument, the required geophysical variable is inferred in some way.

> P3, L34: We have clarified that SEVIRI does not directly observe radiometric height and that CALIOP does not directly observe cloud top height.

Page 3, line 34: quantify what you mean by optically thin, and give the wavelength that you are defining the opacity at.

> P4, L20: Information on the sensitivity of CALIOP's cloud optical depth measurements, as well the wavelength at which optical depth is defined, has been added. Cloud optical depths from SEVIRI are not used in this study. Instead, SEVIRI data are excluded based on cloud optical depths from collocated CALIOP measurements. A discussion of the cloud optical depth thresholds used in previous studies and a detailed discussion of how the threshold was chosen for this studies are given in Sect. 3.2.1.

Page 4, line 14: this makes it sound as if you are using data from one instrument flying on one satellite through the whole period considered here which is not true. Neither is it true that throughout this record the operational SEVIRI instrument was at 0 degrees longitude.

P5, L3: We have clarified that MSG is a series of satellites, as well as the fact that older versions of the satellites, which continue to fly and are located further to the east, are occasionally used to fill gaps in data coverage.

Page 4, line 31: more detail on how this CTP retrieval works is required. For example, the wording 'the data available', is extremely vague.

P6, L1: We have extended the description of the algorithms used for cloud masking and cloud top pressure retrievals in section 2. We have also removed the phrase "and the data available". In some cases, the NWCSAF/MSGv2012 retrieval process changes depending on whether certain variables are missing from the satellite, or model datasets. However, upon further investigation, the CLAAS-2 dataset used in this study only makes retrievals where all such datasets are complete.

Page 5, line 10: as noted in the major comments, more detail on how this is done is required.

P6, L1: We have extended the description of the algorithms used for cloud masking and cloud top pressure retrievals in section 2.

Figure 1: I find this figure, although useful, somewhat misleading as it implies that clouds are ubiquitous. I would suggest another figure or set of panels indicating frequency of cloud occurrence would be beneficial. Similar information is shown later on in the paper but I think it would be good to have it upfront.

P7, L11: We have moved Fig. 7, which shows the frequency of cloud occurrence for the full CLAAS-2 dataset, so that it follows directly from Fig. 1.

Page 6, line 12: while I agree that the diurnal cycle could well be largest here I don't think it necessarily follows from the reasoning given. It could be that the cold clouds are there throughout the day. Figure 1 tells you nothing about this on its own.

P7, L30: We agree with the reviewer that this figure says nothing about the diurnal cycle by itself. We have clarified that this figure is only indicative of areas which potentially have a strong diurnal cycle of convection and that a full analysis of the diurnal cycle (presented later in the paper) will be necessary.

Page 6, line 18: Figure 1 implies a spatial and seasonal pattern in cloud type but it doesn't explicitly show it. Since you state that the cloud retrieval you are using gives cloud type it would be interesting to see if that maps to what you see in Figure 1.

P8, L3: We agree with the reviewer that this would be very interesting. However, since the cloud type information has not been processed by EUMETSAT in the same summarizing way as other cloud properties, this analysis is not currently available and would require substantial additional work, using data not publically available from EUMETSAT at this time. We have added a discussion of the assumed strong seasonal variability in cloud type over Africa due to the cyclic movement of the ITCZ.

Page 6, line 20: I don't think section 2 really gives a quantitative idea of the implications of cloud type and land surface emissivity for the accuracy of CLAAS CTT. There was more of a general discussion in the introduction to be honest.

P8, L7: We have amended the reference to point to the more in depth discussion in the introduction.

Page 6, line 25: When the authors say 'data was processed' are they referring simply to the collocation process? It might be good to state this explicitly.

P8, L11: The reviewer is correct that were referring to the collocation process. This has been clarified.

Page 6, line 30: 'very accurate' is not very scientific.

P8, L16: We have added further details and references regarding the accuracy of CALIOP's measurement of CTH.

Page 7, line 17: define COD as Cloud Optical Depth (and give wavelength).

P4, L19: COD has been previously defined on P6, L26. Information on the wavelength at which COD is measured has been added to P4, L19.

Page 7, line 25: I assume the temporal collation window is centred on the CALIOP observation. Is the variation in the timing of SEVIRI scan lines accounted for? What is the spatial match up criterion?

P9, L16: We have added a paragraph clarifying our collocation methodology to the start of the discussion of the collocation process.

Page 7, line 28: Actually figure 2 suggests that the tightening from 0.3 to 1 does result in a reduced bias while above 1 there is not much change. The authors say this themselves in the next paragraph.

P10, L6: We mean that when viewing a high COD cloud, CALIOP measures the physical cloud top, while SEVIRI observes the radiometric cloud top, which is below the height of the physical cloud top. The application of COD thresholds can remove cases where CALIOP might report the height of a cloud with an optical depth of, say 0.1, while SEVIRI observes the height of thicker cloud below. These thresholds cannot however account for the difference between the physical properties observed by the two instruments. A discussion of the expected size of this difference is provided in Sect. 1.

We thank the reviewer for pointing out that the distinction between the two issues needs to be clarified and have amended our text accordingly.

Page 7, line 30: How is a cloud layer defined? i.e. are the thresholds simply applied to the topmost layer with a COD as diagnosed by the CALIOP product? Is there any change in vertical resolution in this product? I'm not sure whether it would have an impact but if for example the vertical resolution reduces with height above a certain point, the same COD would actually indicate a more diffuse extinction profile within the layer.

P10, L8: The reviewer is correct in their assumption that thresholds are applied to the topmost cloud layer with a COD as diagnosed by the CALIOP product. We have clarified this point in our text.

With regard to the vertical resolution of CALIOP, this does change with height. Between the surface and ~8.2 km, the vertical is 30 m. From ~8.2 km to ~20.2 km it is 60 m. The same COD could therefore indicate different extinction profiles above and below ~8.2 km. While several previous studies (Reuter et al., 2009; Stubenrauch et al., 2010; SAFNWC/MSG, 2012; Kniffka et al., 2013; Benas et al., 2016) have used CALIOP layer optical depth as a threshold to exclude low optical depth clouds when comparing cloud top properties from SEVIRI and CALIOP, we do agree that this may introduce a bias. We have therefore added an explanation of the potential biases introduced by the change in vertical resolution of CALIOP (P9, L14).

Page 8, line 13: what is the 'relatively coarse' resolution in numbers? How does this compare to the CALIOP vertical resolution? How well do both of the meteorological models actually capture low level temperature inversions? Do they persist throughout the night and day?

P6, L1: We have extended the description of the algorithms used for cloud masking and cloud top pressure retrievals from SEVIRI in section 2 and included references relating to the model used.

P9, L1: We have also extended the description of the algorithms used by the CALIOP retrieval and included references relating to the model used.

We have contacted the creators of the SEVIRI and CALIOP datasets, but have been unable to obtain further information on the accuracy of these models with regards to low level temperature inversions.

Page 9, line 7: do the deviations show a Gaussian distribution?

P11, L19: The biases between SEVIRI and CALIOP CTTs (SEVIRI minus CALIOP) show a right-skewed Gaussian distribution (below). In general, positive values in the distribution show biases due to differences between the radiometric and physical CTTs retrieved from the two instruments, while negative values indicate a potential retrieval error. In particular, it seems that the distribution may be skewed slightly towards the negative due to the large number of slightly negative values found in the southeast Atlantic Ocean. A brief discussion of the shape of the distribution has been added to the text.

[Figure]

Figure 5: I would weight these differences by occurrence or at the very least discuss them in the context of Figure 4. Otherwise the eye is drawn to the very strong positive daytime bias over the Sahara when really there are few clouds there. Why are there negative differences over the east of Africa during the daytime (Sudan/Ethiopia)?

P12, L5: we agree with the reviewer that it is important to consider Figure 5 in the context of the occurrence of cloud in different regions. We feel that weighting the biases by occurrence would unhelpfully obscure the magnitude of the biases in some regions. We have however amended our discussion to highlight the importance of considering Figure 5 (now Fig. 6) in the context of Figure 4 (now Fig.5).

Re: East Africa. We also found the negative differences over Sudan/Ethiopia during the daytime puzzling. Unfortunately, we have been unable to find an explanation for this feature and further investigation of this feature is left to future studies.

Page 9, line 28: it may be the colour scale but I would say that while the majority of the ocean shows small biases, there are regions where differences look relatively large. One of these regions is discussed in the next paragraph so perhaps merge these two parts together.

P12, L12: Although there are some regions over the ocean with relatively high bias, these still fall within the range of biases expected due to the difference between the radiometric and physical cloud top heights observed by SEVIRI and CALIOP respectively. We have added a brief discussion of this point to the text.

Page 10, line 20: I understand that the authors have used three month means in order to obtain sufficient data to see a coherent cycle at the pixel scale but do they perhaps worry that by doing this you are losing information about how the phase of any diurnal cycle in CTT might vary through the year? The discussion about producing 'smooth cycles' is rather vague.

P13, L7: We agree with the reviewer that there is a risk that averaging data over multiple months may obscure information about variation in the diurnal cycle throughout the year. We have attempted to balance the need to have enough data for a statistical analysis of the diurnal cycle in regions with sparse data (e.g. over land) with our interest in analyzing annual variation in the diurnal cycle. We believe that by breaking the dataset down into three-monthly means we are able to show the most important features of the variability throughout the year, while retaining sufficient data for a statistical analysis over the majority of the land.

Our discussion of 'smooth' cycles references our attempt to ensure that in every region there were enough CTT retrievals at each hour in the day to produce mean diurnal cycles in which a single data point (e.g. the data point at 19:00 in Fig. 6) represents a mean over at least 1,000 cloud top temperature retrievals for the majority of the area included in this study. Areas for which this was not possible are shaded in Figures 8 and 9.

We have clarified the process by which the diurnal cycle was calculated and replaced the reference to 'smooth cycles'.

Figure 6: Can the authors provide some idea of the range of values that comprise each hourly mean please, perhaps using quartiles or SDs if the distribution is Gaussian?

Figure 7: The range of values that comprise each hourly mean are now shown in Figure 7, which now includes box plots showing quartiles for each hour. Additional text discussing these values has also been added.

Page 10, line 33 (and in other sections focused on the Sahara): I am a bit bemused about the emphasis on the Sahara in the latter part of the paper. As is shown, there is very little cloud being detected there (as one might anticipate) and I am not sure that I would expect too much of what is there to be deep convective in nature (at least north of the inter-tropical front). Hence why should we expect a marked diurnal cycle? Moreover, when what is happening there is analysed the statistics will be poor.

We agree that few clouds and no marked diurnal cycle are expected in the Sahara. The reason for our focus on this region is the large negative biases seen in the daytime, as well as the related large difference between day and nighttime biases is this area. Although this is of course based on sparse data, the spatial uniformity of the bias in this region is itself interesting. Our purpose in discussing the region is to show that these biases lead to a large, unphysical, diurnal cycle being observed in this region, as well as to show that the large negative daytime biases in CTT in this region may indicate a potential issue with the SEVIRI retrieval algorithm.

Page 12, line 33: I agree that vegetated Central Africa will typically have a lower surface albedo than the Sahara but I don't see why this would cause the Sahara to heat up more quickly after sunrise (I would actually expect the opposite based on albedo alone) or how this would produce lower, warmer clouds. Please explain.

P15, L23: We thank the reviewer for their comment on this point. We revised our arguments and agree with the reviewer that the facts do not support this claim. For this reason, we have removed this discussion and leave the investigation of this feature to future studies. We have retained a discussion of the differences in the diurnal cycle between these two regions, and highlighted the lack of a robust explanation for the difference.

Page 13, line 22: 'Additional biases' – this sentence is very vague. What retrieval errors are being referred to?

We do not claim to have isolated the cause of any retrieval errors in the CLAAS-2 dataset. We simply mean to state that while the majority of the biases described in this paper can potentially be explained by either the expected difference in CTT due to the fact that SEVIRI observes the radiometric and CALIOP measures the physical cloud top, or due to differences in the approach to dealing with atmospheric inversions between the two datasets, there of course may be a few areas where neither of these explanations apply. It may be that in these cases, retrieval errors in the

CLAAS-2 dataset are contributing to the observed biases. It is however, beyond the scope of this paper to identify the cause of further retrieval biases.

P16, L17: We have amended our text to clarify our meaning.

---

## Author Comment (AC4) · 27 Feb 2017

The comment was uploaded in the form of a supplement:
http://www.atmos-chem-phys-discuss.net/acp-2016-878/acp-2016-878-AC4-supplement.pdf

---

## Referee Report (RR1)

Second Review of 'Evaluating the diurnal cycle in cloud top temperature from SEVIRI' by Taylor et al.

The authors have made a good effort to address the points raised in the initial review. I particularly appreciate the additional information concerning the data algorithms and processing used to perform the study. There are one or two minor issues that still remain but I think these can be fixed quickly without the need for a subsequent review.

NB. Line numbers below refer to the tracked changes version of the manuscript.

**Introduction:**

In their efforts to make this more general some of the consistency in describing cloud regimes has been lost – for example the majority of the references in line 32 on page 2 are, I think, referring to a phasing in convective cloud, not all cloud types, a suspicion reinforced by the next sentence. Please check carefully that simply removing the word 'convective' or 'convection' is appropriate on each occasion that this has been done.

Just for info, the implications of Pearson's two studies is that it is the scale at which a convective parameterization scheme is employed **and** the mechanism used to represent convection, rather than spatial resolution per se, that is key for improving the representation of the diurnal evolution and growth of tropical convective systems. This is a little contrary to what has been written.

Page 4, line 5, SEVIRI will underestimate CTH and overestimate CTP. It can't do the same thing for both ☺.

Page 8, line 21, …(Benas et al., 2016),…

Page 16, line 17. Appreciate the effort to clarify what is meant but the sentence is weak. Possibly better: 'We believe that biases that fall outside of both the 3-20 K range and the region of subsidence in the southeast Atlantic Ocean are most likely the result of other, as yet undiagnosed, errors in the SEVIRI retrievals. However, it is not….'

Appendix A:

I think your 60 minute window should be +/- 30 minutes not 15? Actually, the first para of the appendix is a little repetitive with the additional information. Although the authors state that they find the insensitivity surprising they actually provide several sensible reasons why it may, on further reflection, not be. It's a very minor point but perhaps relate the final para back to the initial expectation? Suggest:

'The insensitivity of the calculated bias in SEVIRI CTT to a change in the collocation window used for matching to CALIOP may initially seem surprising. We collocated SEVIRI and CALIOP CTTs, for the full year of 2007, using both 60 minute (+/- 30 minutes of CALIOP overpass) and a 15 minute (+/- 7.5 minutes of CALIOP overpass) collocation windows. This amounts to an extra 22.5 minutes between CALIOP and SEVIRI retrievals in the 60 minute window case, as compared to the 15 minute case.'

And (less necessary)…

'On reflection, there are many reasons….'

---

## Author Response (AR2)

**Second Review of 'Evaluating the diurnal cycle in cloud top temperature from SEVIRI' by Taylor et al.**

The authors have made a good effort to address the points raised in the initial review. I particularly appreciate the additional information concerning the data algorithms and processing used to perform the study. There are one or two minor issues that still remain but I think these can be fixed quickly without the need for a subsequent review.

> **N.B.** Line numbers below refer to the tracked changes version of the manuscript.

**Introduction:**
In their efforts to make this more general some of the consistency in describing cloud regimes has been lost – for example the majority of the references in line 32 on page 2 are, I think, referring to a phasing in convective cloud, not all cloud types, a suspicion reinforced by the next sentence. Please check carefully that simply removing the word 'convective' or 'convection' is appropriate on each occasion that this has been done.
> P2, L23: We thank the reviewer for raising this point and have clarified that the paragraph in question refers to convective cloud only. We have also checked the rest of the section to ensure that descriptions of cloud types are consistent.

Just for info, the implications of Pearson's two studies is that it is the scale at which a convective parameterization scheme is employed and the mechanism used to represent convection, rather than spatial resolution per se, that is key for improving the representation of the diurnal evolution and growth of tropical convective systems. This is a little contrary to what has been written.
> P3, L4: We have clarified the implications of the Pearson papers.

Page 4, line 5, SEVIRI will underestimate CTH and overestimate CTP. It can't do the same thing for both.
> P3, L30: we have amended this line as suggested.

Page 8, line 21, ...(Benas et al., 2016),...
> P7, L20: we have changed the citation format.

Page 16, line 17. Appreciate the effort to clarify what is meant but the sentence is weak. Possibly better: 'We believe that biases that fall outside of both the 3-20 K range and the region of subsidence in the southeast Atlantic Ocean are most likely the result of other, as yet undiagnosed, errors in the SEVIRI retrievals. However, it is not....'
> P15, L6: we have changed the sentence, as suggested above.

Appendix A:
I think your 60 minute window should be +/- 30 minutes not 15? Actually, the first para of the appendix is a little repetitive with the additional information. Although the authors state that they find the insensitivity surprising they actually provide several sensible reasons why it may, on further reflection, not be. It's a very minor point but perhaps relate the final para back to the initial expectation? Suggest:
'The insensitivity of the calculated bias in SEVIRI CTT to a change in the collocation window used for matching to CALIOP may initially seem surprising. We collocated SEVIRI and CALIOP CTTs, for the full year of 2007, using both 60 minute (+/- 30 minutes of CALIOP overpass) and a 15 minute (+/- 7.5 minutes of CALIOP overpass) collocation windows. This amounts to an extra 22.5 minutes between CALIOP and SEVIRI retrievals in the 60 minute window case, as compared to the 15 minute case.'

P15, L32 we have amended this paragraph as suggested.

And (less necessary)… 'On reflection, there are many reasons....'
P16, L17: we have rephrased this sentence.

[revised manuscript text omitted]